# SURGE: Surface Regularized Geometry Estimation from a Single Image

**Peng Wang**[1]  **Xiaohui Shen**[2]  **Bryan Russell**[2]  **Scott Cohen**[2]  **Brian Price**[2]  **Alan Yuille**[3]

[1]University of California, Los Angeles  [2]Adobe Research  [3]Johns Hopkins University

## Abstract

This paper introduces an approach to regularize 2.5D surface normal and depth predictions at each pixel given a single input image. The approach infers and reasons about the underlying 3D planar surfaces depicted in the image to snap predicted normals and depths to inferred planar surfaces, all while maintaining fine detail within objects. Our approach comprises two components: (i) a four-stream convolutional neural network (CNN) where depths, surface normals, and likelihoods of planar region and planar boundary are predicted at each pixel, followed by (ii) a dense conditional random field (DCRF) that integrates the four predictions such that the normals and depths are compatible with each other and regularized by the planar region and planar boundary information. The DCRF is formulated such that gradients can be passed to the surface normal and depth CNNs via backpropagation. In addition, we propose new planar-wise metrics to evaluate geometry consistency within planar surfaces, which are more tightly related to dependent 3D editing applications. We show that our regularization yields a 30% relative improvement in planar consistency on the NYU v2 dataset [24].

## 1  Introduction

Recent efforts to estimate the 2.5D layout of a depicted scene from a single image, such as per-pixel depths and surface normals, have yielded high-quality outputs respecting both the global scene layout and fine object detail [2, 6, 7, 29]. Upon closer inspection, however, the predicted depths and normals may fail to be consistent with the underlying surface geometry. For example, consider the depth and normal predictions from the contemporary approach of Eigen and Fergus [6] shown in Figure 1 (b) (Before DCRF). Notice the significant distortion in the predicted depth corresponding to the depicted planar surfaces, such as the back wall and cabinet. We argue that such distortion arises from the fact that the 2.5D predictions (i) are made independently per pixel from appearance information alone, and (ii) do not explicitly take into account the underlying surface geometry. When 3D geometry has been used, e.g., [29], it often consists of a boxy room layout constraint, which may be too coarse and fail to account for local planar regions that do not adhere to the box constraint. Moreover, when multiple 2.5D predictions are made (e.g., depth and normals), they are not explicitly enforced to agree with each other.

To overcome the above issues, we introduce an approach to identify depicted 3D planar regions in the image along with their spatial extent, and to leverage such planar regions to regularize the depth and surface normal outputs. We formulate our approach as a four-stream convolutional neural network (CNN), followed by a dense conditional random field (DCRF). The four-stream CNN independently predicts at each pixel the surface normal, depth, and likelihoods of planar region and planar boundary. The four cues are integrated into a DCRF, which encourages the output depths and normals to align with the inferred 3D planar surfaces while maintaining fine detail within objects. Furthermore, the output depths and normals are explicitly encouraged to agree with each other.

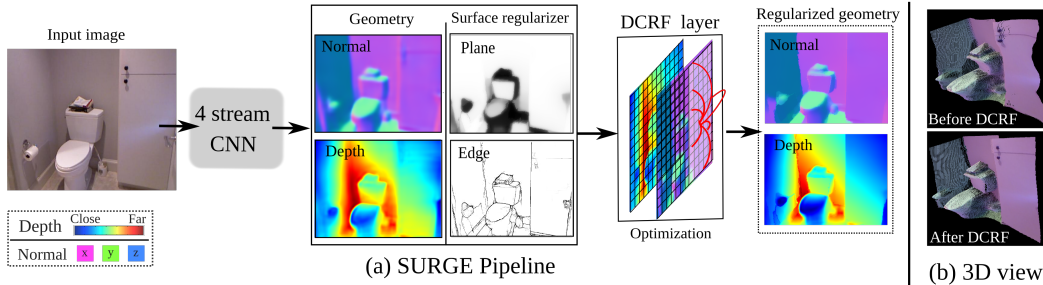

Figure 1: Framework of SURGE system. (a) We induce surface regularization in geometry estimation though DCRF, and enable joint learning with CNN, which largely improves the visual quality (b).

We show that our DCRF is differentiable with respect to depth and surface normals, and allows back-propagation to the depth and normal CNNs during training. We demonstrate that the proposed approach shows relative improvement over the base CNNs for both depth and surface normal prediction on the NYU v2 dataset using the standard evaluation criteria, and is significantly better when evaluated using our proposed plane-wise criteria.

## 2 Related work

From a single image, traditional geometry estimation approaches rely on extracting visual primitives such as vanishing points and lines [10] or abstract the scenes with major plane and box representations [22, 26]. Those methods can only obtain sparse geometry representations, and some of them require certain assumptions (*e.g.* Manhattan world).

With the advance of deep neural networks and their strong feature representation, dense geometry, i.e., pixel-wise depth and normal maps, can be readily estimated from a single image [7]. Long-range context and semantic cues are also incorporated in later works to refine the dense prediction by combining the networks with conditional random fields (CRF) [19, 20, 28, 29]. Most recently, Eigen and Fergus [6] further integrate depth and normal estimation into a large multi-scale network structure, which significantly improves the geometry estimation accuracy. Nevertheless, the output of the networks still lacks regularization over planar surfaces due to the adoption of pixel-wise loss functions during network training, resulting in unsatisfactory experience in 3D image editing applications.

For inducing non-local regularization, DCRF has been commonly used in various computer vision problems such as semantic segmentation [5, 32], optical flow [16] and stereo [3]. However, the features for the affinity term are mostly simple ones such as color and location. In contrast, we have designed a unique planar surface affinity term and a novel compatibility term to enable 3D planar regularization over geometry estimation.

Finally, there is also a rich literature in 3D reconstruction from RGBD images [8, 12, 24, 25, 30], where planar surfaces are usually inferred. However, they all assume that the depth data have been acquired. To the best of our knowledge, we are the first to explore using planar surface information to regularize dense geometry estimation by only using the information of a single RGB image.

## 3 Overview

Fig. 1 illustrates our approach. An input image is passed through a four-stream convolutional neural network (CNN) that predicts at each pixel a surface normal, depth value, and whether the pixel belongs to a planar surface or edge (i.e., edge separating different planar surfaces or semantic regions), along with their prediction confidences. We build on existing CNNs [6, 31] to produce the four maps.

While the CNNs for surface normals and depths produce high-fidelity outputs, they do not explicitly enforce their predictions to agree with depicted planar regions. To address this, we propose a fully-connected dense conditional random field (DCRF) that reasons over the CNN outputs to regularize the surface normals and depths. The DCRF jointly aligns the surface normals and depths to individual planar surfaces derived from the edge and planar surface maps, all while preserving fine detail within objects. Our DCRF leverages the advantages of previous fully-connected CRFs [15] in terms of both its non-local connectivity, which allows propagation of information across an entire planar surface, and efficiency during inference. We present our DCRF formulation in Section 4, followed by our algorithm for joint learning and inference within a CNN in Section 5.

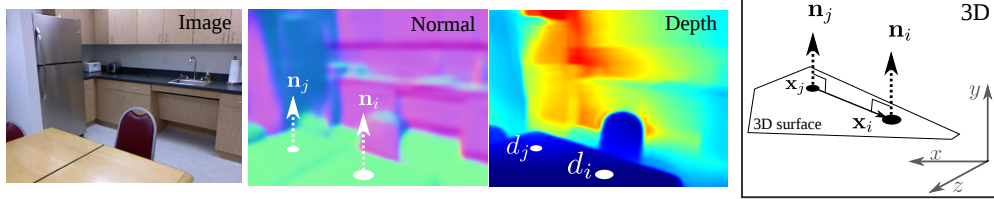

Figure 2: The orthogonal compatibility constraint inside the DCRF. We recover 3d points from the depth map and require the difference vector to be perpendicular to the normal predictions.

## 4 DCRF for Surface Regularized Geometry Estimation

In this section, we present our DCRF that incorporates plane and edge predictions for depth and surface normal regularization. Specifically, the field of variables we optimize are depths, $\mathbf{D} = \{d_i\}_{i=1}^K$, where $K$ is number of the pixels, and normals, $\mathbf{N} = \{\mathbf{n}_i\}_{i=1}^K$, where $\mathbf{n}_i = [n_{ix}, n_{iy}, n_{iz}]^T$ indicates the 3D normal direction.

In addition, as stated in the overview (Sec. 3), we have four types of information from the CNN predictions, namely a predicted normal map $\mathbf{N}_o = \{\mathbf{n}_i^o\}_{i=1}^K$, a depth map $\mathbf{D}_o = \{d_i\}_{i=1}^K$, a plane probability map $\mathbf{P}_o$ and edge predictions $\mathbf{E}_o$. Following the general form of DCRF [16], our problem can be formulated as,

$$\min_{\mathbf{N},\mathbf{D}} \sum_i \psi_u(\mathbf{n}_i, d_i | \mathbf{N}_o, \mathbf{D}_o) + \lambda \sum_{i,j,i \neq j} \psi_r(\mathbf{n}_i, \mathbf{n}_j, d_i, d_j | \mathbf{P}_o, \mathbf{E}_o) \text{ with } \|\mathbf{n}_i\|_2 = 1 \quad (1)$$

where $\psi_u(\cdot)$ is a unary term encouraging the optimized surface normals $\mathbf{n}_i$ and depths $d_i$ to be close to the outputs $\mathbf{n}_i^o$ and $d_i^o$ from the networks. $\psi_r(\cdot, \cdot)$ is a pairwise fully connected regularization term depending on the information from the plane map $\mathbf{P}_o$ and edge map $\mathbf{E}_o$, where we seek to encourage consistency of surface normals and depths within planar regions with the underlying depicted 3D planar surfaces. Also, we constrain the normal predictions to have unit length. Specifically, the definition of unary and pairwise in our model are presented as follows.

### 4.1 Unary terms

Motivated by Monte Carlo dropout [27], we notice that when forward propagating multiple times with dropout, the CNN predictions have different variations across different pixels, indicating the prediction uncertainty. Based on the prediction variance from the normal and depth networks, we are able to obtain pixel-wise confidence values $w_i^n$ and $w_i^d$ for normal and depth predictions. We leverage such information to DCRF inference by trusting the predictions with higher confidence while regularizing more over ones with low confidence. By integrating the confidence values, our unary term is defined as,

$$\psi_u(\mathbf{n}_i, d_i | \mathbf{N}_o, \mathbf{D}_o) = \frac{1}{2} w_i^n \psi_n(\mathbf{n}_i | \mathbf{n}_o) + \frac{1}{2} w_i^d \psi_d(d_i | d_o), \quad (2)$$

where $\psi_n(\mathbf{n}_i | \mathbf{n}_o) = 1 - \mathbf{n}_i \cdot \mathbf{n}_i^o$ is the cosine distance between the input and output surface normals, and $\psi_d(d_i | d_o) = (d_i - d_i^o)^2$ is the is the squared difference between input and output depth.

### 4.2 Pairwise term for regularization.

We follow the convention of DCRF with Gibbs energy [17] for pairwise designing, but also bring in the confidence value of each pixel as described in Sec. 4.1. Formally, it is defined as,

$$\psi_r(\mathbf{n}_i, \mathbf{n}_j, d_i, d_j | \mathbf{P}_o, \mathbf{E}_o) = \left( w_{i,j}^n \mu_n(\mathbf{n}_i, \mathbf{n}_j) + w_{i,j}^d \mu_d(d_i, d_j, \mathbf{n}_i, \mathbf{n}_j) \right) A_{i,j}(\mathbf{P}_o, \mathbf{E}_o),$$

$$\text{where,} \quad w_{i,j}^n = \frac{1}{2}(w_i^n + w_j^n), w_{i,j}^d = \frac{1}{2}(w_i^d + w_j^d) \quad (3)$$

Here, $\mathbf{A}_{i,j}$ is a *pairwise planar affinity* indicating whether pixel locations $i$ and $j$ belong to the same planar surface derived from the inferred edge and planar surface maps. $\mu_n()$ and $\mu_d()$ regularize the output surface normals and depths to be aligned inside the underlying 3D plane. Here, we use simplified notations, i.e. $\mathbf{A}_{i,j}$, $\mu_n()$ and $\mu_d()$ for the corresponding terms.

For the compatibility $\mu_n()$ of surface normals, we use the same function as $\psi_n()$ in Eqn. (2), which measures the cosine distance between $\mathbf{n}_i$ and $\mathbf{n}_j$. For depths, we design an *orthogonal compatibility* function $\mu_d()$ which encourages the normals and depths of each adjacent pixel pair to be consistent and aligned within a 3D planar surface. Next we define $\mu_d()$ and $\mathbf{A}_{i,j}$.

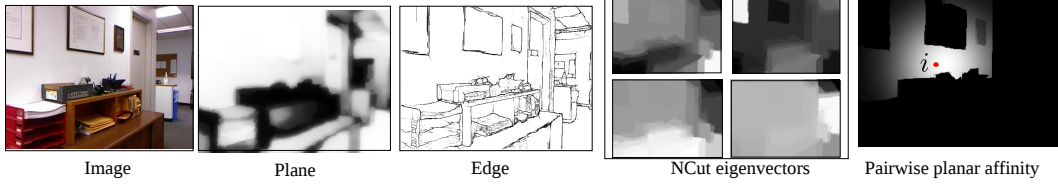

| Image | Plane | Edge | NCut eigenvectors | Pairwise planar affinity |

Figure 3: Pairwise surface affinity from the plane and edge predictions with computed Ncut features. We highlight the computed affinity w.r.t. pixel $i$ (red dot).

**Orthogonal compatibility:** In principle, when two pixels fall in the same plane, the vector connecting their corresponding 3D world coordinates should be perpendicular to their normal directions, as illustrated in Fig. 2. Formally, this orthogonality constraint can be formulated as,

$$\mu_d(d_i, d_j, \mathbf{n}_i, \mathbf{n}_j) = \frac{1}{2}\left(\mathbf{n}_i \cdot (\mathbf{x}_i - \mathbf{x}_j)\right)^2 + \frac{1}{2}\left(\mathbf{n}_j \cdot (\mathbf{x}_i - \mathbf{x}_j)\right)^2, \text{ with } \mathbf{x}_i = d_i \mathbf{K}^{-1}\mathbf{p}_i. \quad (4)$$

Here $\mathbf{x}_i$ is the 3D world coordinate back projected by 2D pixel coordinate $\mathbf{p}_i$ (written in homogeneous coordinates), given the camera calibration matrix $\mathbf{K}$ and depth value $d_i$. This compatibility encourages consistency between depth and normals.

**Pairwise planar affinity:** As noted in Eqn. (3), the planar affinity is used to determine whether pixels $i$ and $j$ belong to the same planar surface from the information of plane and edge. Here $\mathbf{P}_o$ helps to check whether two pixels are both inside planar regions, and $\mathbf{E}_o$ helps to determine whether the two pixels belong to the same planar surface. Here, for efficiency, we chose the form of Gaussian bilateral affinity to represent such information since it has been successfully adopted by many previous works with efficient inference, e.g. in discrete label space for semantic segmentation [5] or in continuous label space for edge-aware smoothing [3, 16]. Specifically, following the form of bilateral filters, our planar surface affinity is defined as,

$$\mathbf{A}_{i,j}(\mathbf{P}_o, \mathbf{E}_o) = p_i p_j \left(\omega_1 \kappa\left(\mathbf{f}_i, \mathbf{f}_j; \theta_\alpha\right) \kappa\left(\mathbf{c}_i, \mathbf{c}_j; \theta_\beta\right) + \omega_2 \kappa\left(\mathbf{c}_i, \mathbf{c}_j; \theta_\gamma\right)\right), \quad (5)$$

where $\kappa(\mathbf{z}_i, \mathbf{z}_j; \theta) = \exp\left(-\frac{1}{2\theta^2}\|\mathbf{z}_i - \mathbf{z}_j\|^2\right)$ is a Gaussian RBF kernel. $p_i$ is the predicted value from the planar map $\mathbf{P}_o$ at pixel $i$. $p_i p_j$ indicates that the regularization is activated when both $i, j$ are inside planar regions with high probability. $\mathbf{f}_i$ is the appearance feature derived from the edge map $\mathbf{E}_o$, $\mathbf{c}_i$ is the 2D coordinate of pixel $i$ on image. $\omega_1, \omega_2, \theta_\alpha, \theta_\beta, \theta_\gamma$ are parameters.

To transform the pairwise similarity derived from the edge map to the feature representation $\mathbf{f}$ for efficient computing, we borrow the idea from the Normalized Cut (NCut) for segmentation [14, 23], where we can first generate an affinity matrix between pixels using intervening contour [23], and perform normalized cut. We select the top 6 resultant eigenvectors as our feature $\mathbf{f}$. . A transformation from edge to the planar affinity using the eigenvectors is shown in Fig. 3. As can be seen from the affinity map, the NCut features are effective to determine whether two pixels lie in the same planar surface where the regularization can be performed.

## 5 Optimization

Given the formulation in Sec. 4, we first discuss the fast inference implementation for DCRF, and then present the algorithm of joint training with CNNs through back-propagation.

### 5.1 Inference

To optimize the objective function defined in Eqn.(1), we use mean-field approximation for fast inference as used in the optimization of DCRF [15]. In addition, we chose to use coordinate descent to sequentially optimize normals and depth. When optimizing normals, for simplicity and efficiency, we do not consider the term of $\mu_d()$ in Eqn.(3), yielding the updating for pixel $i$ at iteration $t$ as,

$$\mathbf{n}_i^{(t)} \leftarrow \frac{1}{2} w_i^n \mathbf{n}_i^o + \frac{\lambda}{2} \sum_{j, j \neq i} w_j^n \mathbf{n}_j^{(t-1)} \mathbf{A}_{i,j}, \quad \mathbf{n}_i^{(t)} \leftarrow \mathbf{n}_i^{(t)} / \|\mathbf{n}_i^{(t)}\|_2, \quad (6)$$

which is equivalent to first performing a dense bilateral filtering [4] with our pairwise planar affinity term $\mathbf{A}_{i,j}$ for the predicted normal map, and then applying L2 normalization.

Given the optimized normal information, we further optimize depth values. Similar to normals, after performing mean-field approximation, the inferred updating equation for depth at iteration $t$ is,

$$d_i^{(t)} \leftarrow \frac{1}{\nu_i}\left(w_i^d d_i^o + \lambda(\mathbf{n}_i \cdot \mathbf{p}_i) \sum_{j, j \neq i} \mathbf{A}_{i,j} w_j^d d_j^{(t-1)} (\mathbf{n}_j \cdot \mathbf{p}_j)\right) \quad (7)$$

where $\nu_i = w_i^d + \lambda(\mathbf{n}_i \cdot \mathbf{p}_i)\left(\mathbf{p}_i \cdot \sum_{j,j \neq i} \mathbf{A}_{i,j}w_j^d\mathbf{n}_j\right)$, Since the graph is densely connected, previous work [16] indicates that only a few ($<$10) iterations are need to achieve reasonable performance. In practice we found that 5 iterations for normal inference and 2 iterations for depth inference yielded reasonable results.

## 5.2 Joint training of CNN and DCRF

We further implement the DCRF inference as a trainable layer as in [32] by considering the inference as feedforward process, to enable joint training together with the normal and depth neural networks. This makes the planar surface information able to be back-propagated to the neural networks and further refine their output. We describe the gradients back-propagated to the two networks respectively.

**Back-propagation to the normal network.** Suppose the gradient of normal passed from the upper layer after DCRF for pixel $i$ is $\nabla_f(\mathbf{n}_i)$, which is a 3x1 vector. We now back-propagate it first through the L2 normalization using the equation of $\nabla_{L2}(\mathbf{n}_i) = (\mathbf{I}/\|\mathbf{n}_i\| - \mathbf{n}_i\mathbf{n}_i^T/\|\mathbf{n}_i\|^3)\nabla_f(\mathbf{n}_i)$, and then back-propagate through the mean-field approximation in Eqn. (6) as,

$$\frac{\partial L(\mathbf{N})}{\partial \mathbf{n}_i} = \frac{\nabla_{L2}(\mathbf{n}_i)}{2} + \frac{\lambda}{2}\sum_{j,j \neq i} \mathbf{A}_{j,i}\nabla_{L2}(\mathbf{n}_j), \tag{8}$$

where $L(\mathbf{N})$ is the loss from normal predictions after DCRF, $\mathbf{I}$ is the identity matrix.

**Back-propagation to the depth network.** Similarly for depth, suppose the gradient from the upper layer is $\nabla_f(d_i)$, the depth gradient for back-propagation through Eqn. 7 can be inferred as,

$$\frac{\partial L(\mathbf{D})}{\partial d_i} = \frac{1}{\nu_i}\nabla_f(d_i) + \lambda(\mathbf{n}_i \cdot \mathbf{p}_i)\sum_{j,j \neq i}\frac{1}{\nu_j}\mathbf{A}_{j,i}(\mathbf{n}_j \cdot \mathbf{p}_j)\nabla_f(d_j) \tag{9}$$

where $L(\mathbf{D})$ is the loss from depth predictions after DCRF.

Note that during back propagation for both surface normals and depths we drop the confidences $w$ since using it during training will make the process very complicated and inefficient. We adopt the same surface normal and depth loss function as in [6] during joint training. It is possible to also back propagate the gradients of the depth values to the normal network via the surface normal and depth compatibility in Eqn. (4). However, this involves the depth values from all the pixels within the same plane, which may be intractable and cause difficulty during joint learning. We therefore chose not to back propagate through the compatibility in our current implementation and leave it to future work.

## 6 Implementation details for DCRF

To predict the input surface normals and depths, we build on the publicly-available implementation from Eigen and Fergus [6], which is at or near state of the art for both tasks. We compute prediction confidences for the surface normals and depths using Monte Carlo dropout [27]. Specifically, we forward propagate through the network 10 times with dropout during testing, and compute the prediction variance $v_i$ at each pixel. The predictions with larger variance $v_i$ are considered less stable, so we set the confidence as $w_i = \exp(-v_i/\sigma.^2)$. We empirically set $\sigma_n = 0.1$ for normals prediction and $\sigma_d = 0.15$ for depth prediction to produce reasonable confidence values.

Specifically, for prediction the plane map $\mathbf{P}_o$, we adopt a semantic segmentation network structure similar to the Deeplab [5] network but with multi-scale output as the FCN [21]. The training is formulated as a pixel-wise two-class classification problem (planar *vs.* non-planar). The output of the network is hereby a plane probability map $\mathbf{P}_o$ where $p_i$ at pixel $i$ indicates the probability of pixel $i$ belonging to a planar surface. The edge map $\mathbf{E}_o$ indicates the plane boundaries. During training, the ground-truth edges are extracted from the corresponding ground-truth depth and normal maps, and refined by semantic annotations when available (see Fig.4 for an example). We then adopt the recent Holistic-nested Edge Detector (HED) network [31] for training. In addition, we augment the network by adding predicted depth and normal maps as another 4-channel input to improve recall, which is very important for our regularization since missing edges could mistakenly merge two planes and propagate errors during the message passing.

For the surface bilateral filter in Eqn. (5), we set the parameters $\theta_\alpha = 0.1, \theta_\beta = 50, \theta_\gamma = 3, \omega_1 = 1, \omega_2 = 0.3$, and set the $\lambda = 2$ in Eqn.(1) through a grid search over a validation set from [9]. The four types of inputs to the DCRF are aligned and resized to 294x218 by matching the network output of [6]. During the joint training of DCRF and CNNs, we fix the parameters and fine-tune the network

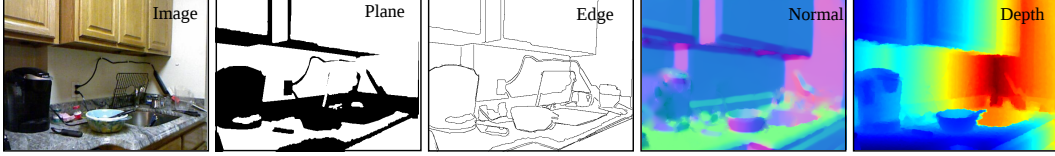

Figure 4: Four types of ground-truth from the NYU dataset that are used in our algorithm.

based on the weights pre-trained from [6], with the 795 training images, and use the same loss functions and learning rates as in their depth and normal networks respectively.

Due to limited space, the detailed edge and plane network structures, the learning and inference times and visualization of confidence values are presented in the supplementary materials.

# 7 Experiments

We perform all our experiments on the NYU v2 dataset [24]. It contains 1449 images with size of 640×480, which is split to 795 training images and 654 testing images. Each image has an aligned ground-truth depth map and a manually annotated semantic category map. In additional, we use the ground-truth surface normals generated by [18] from depth maps. We further use the official NYU toolbox[1] to extract planar surfaces from the ground-truth depth and refine them with the semantic annotations, from which a binary ground-truth plane map and an edge map are obtained. The details of generating plane and edge ground-truth are elaborated in supplementary materials. Fig. 4 shows the produced four types of ground-truth maps for our learning and evaluation.

We implemented all our algorithms based on Caffe [13], including DCRF inference and learning, which are adapted from the implementation in [1, 32].

**Evaluation setup.** In the evaluation, we first compare the normals and depths generated by different baselines and components over the ground truth planar regions, since these are the regions where we are trying to improve, which are most important for 3D editing applications. We evaluated over the valid 561x427 area following the convention in [18, 20]. We also perform evaluation over the ground truth edge area showing that our results preserve better geometry details. Finally, we show the improvement achieved by our algorithm over the entire image region.

We compare our results against the recent work Eigen et.al [6] since it is or is near state-of-the-art for both depth and normal. In practice, we use their published results and models for comparison. In addition, we implemented a baseline method for hard planar regularization, in which the planar surfaces are explicitly extracted from the network predictions. The normal and depth values within each plane are then used to fit the plane parameters, from which the regularized normal and depth values are obtained. We refer to this baseline as "Post-Proc.". For normal prediction, we implemented another baseline in which a basic Bilateral filter based on the RGB image is used to smooth the normal map.

In terms of the evaluation criteria, we first adopt the pixel-wise evaluation criteria commonly used by previous works [6, 28]. However, as mentioned in [11], such metrics mainly evaluate pixel-wise depth and normal offsets, but do not well reflect the quality of reconstructed structures over edges and planar surfaces. Thus, we further propose plane-wise metrics that evaluate the consistency of the predictions inside a ground truth planar region. In the following, we first present evaluations for normal prediction, and then report the results of depth estimation.

**Surface normal criteria.** For pixel-wise evaluation, we use the same metrics used in [6].

For plane-wise evaluation, given a set of ground truth planar regions $\{\mathbf{P}_j^*\}_{j=1}^{N_P}$, we propose two metrics to evaluate the consistency of normal prediction within the planar regions,

1. Degree variation (var.): It measures the overall planarity inside a plane, and defined as, $\frac{1}{N_P} \sum_j \frac{1}{|\mathbf{P}_j^*|} \sum_{i \in \mathbf{P}_j^*} \delta(\mathbf{n}_i, \overline{\mathbf{n}}_j)$, where $\delta(\mathbf{n}_i, \mathbf{n}_j) = \mathrm{acos}(\mathbf{n}_i \cdot \mathbf{n}_j)$ which is the degree difference between two normals, $\overline{\mathbf{n}}_j$ is the normal mean of the prediction inside $\mathbf{P}_j^*$.

2. First-order degree gradient (grad.): It measures the smoothness of the normal transition inside a planar region. Formally, it is defined as, $\frac{1}{N_P} \sum_j \frac{1}{|\mathbf{P}_j^*|} \sum_{i \in \mathbf{P}_j^*} (\delta(\mathbf{n}_i, \mathbf{n}_{hi}) + \delta(\mathbf{n}_i, \mathbf{n}_{vi}))$, where $\mathbf{n}_{hi}, \mathbf{n}_{vi}$ are normals of right and bottom neighbor pixels of $i$.

| | Pixel-wise (Over planar region) | | | | | Plane-wise | |
| | Lower the better | | Higher the better | | | Lower the better | |
| Evaluation over the planar regions | | | | | | | |
| Method | mean | median | 11.25° | 22.5° | 30° | var. | grad. |
|---|---|---|---|---|---|---|---|
| Eigen-VGG [6] | 14.5425 | 8.9735 | 59.00 | 80.85 | 87.38 | 9.1534 | 1.1112 |
| RGB-Bilateral | 14.4665 | 8.9439 | 59.12 | 80.86 | 87.41 | 8.6454 | 1.1735 |
| Post-Proc. | 14.8154 | 8.6971 | 59.85 | 80.52 | 86.67 | 7.2753 | 0.9882 |
| Eigen-VGG (JT) | 14.4978 | 8.9371 | 59.12 | 80.90 | 87.43 | 8.9601 | 1.0795 |
| DCRF | 14.1934 | 8.8697 | 59.27 | 81.08 | 87.77 | 6.9688 | 0.7441 |
| DCRF (JT) | 14.2055 | 8.8696 | 59.34 | 81.13 | 87.78 | 6.8866, | 0.7302 |
| DCRF-conf | **13.9732** | 8.5320 | 60.89 | 81.87 | **88.09** | 6.8212 | 0.7407 |
| DCRF-conf (JT) | 13.9763 | **8.2535** | **62.20** | **82.35** | 88.08 | **6.3939** | **0.6858** |
| Oracle | 13.5804 | 8.1671 | 62.83 | 83.16 | 88.85 | 4.9199 | 0.5923 |
| Eigen-VGG [6] | **23.4141** | 18.3288 | 30.90 | 58.91 | **71.43** | Edge | |
| DCRF-conf (JT) | 23.4694 | **17.6804** | **33.63** | **59.53** | 71.03 | | |
| Eigen-VGG [6] | 20.9322 | 13.2214 | 44.43 | 67.25 | 75.83 | Image | |
| DCRF-conf (JT) | **20.6093** | **12.1704** | **47.29** | **68.92** | **76.64** | | |

Table 1: Normal accuracy comparison over the NYU v2 dataset. We compare our final results (DCRF-conf (JT)) against various baselines over ground truth planar regions at upper part, where JT means joint training CNN and DCRF as presented in Sec. 5.2. We list additional comparison over the edge and full image region at lower part.

**Evaluation on surface normal estimation.** In upper part of Tab. 1, we show the comparison results. The first line, i.e. Eigen-VGG, is the result from [6] with VGG net, which serves as our baseline. The simple RGB-Bilateral filtering can only slightly improve the network output since it does not contain any planar surface information during the smoothing. The hard regularization over planar regions ("Post-Proc.") can improve the plane-wise consistency since hard constraints are enforced in each plane, but it also brings strong artifacts and suffers significant decrease in pixel-wise accuracy metrics. Our "DCRF" can bring improvement on both pixel-wise and plane-wise metrics, while integrating network prediction confidence further makes the DCRF inference achieve much better results. Specifically, using "DCRF-conf", the plane-wise error metric var. drops from 9.15 produced by the network to 6.8. It demonstrates that our non-local planar surface regularization does help the predictions especially for the consistency inside planar regions.

We also show the benefits from the joint training of DCRF and CNN. "Eigen-VGG (JT)" denotes the output of the CNN after joint training, which shows better results than the original network. It indicates that regularization using DCRF for training also improves the network. By using the joint trained CNN and DCRF ("DCRF (JT)"), we observe additional improvement over that from "DCRF". Finally, by combining the confidence from joint trained CNN, our final outputs ("DCRF-conf (JT)") achieve the best results over all the compared methods. In addition, we also use ground-truth plane and edge map to regularize the normal output("Oracle") to get an upper bound when the planar surface information is perfect. We can see our final results are in fact quite close to "Oracle", demonstrating the high quality of our plane and edge prediction.

In the bottom part of Tab. 1, we show the evaluation over edge areas (rows marked by "Edge") as well as on the entire images (marked by "Image"). The edge areas are obtained by dilating the ground truth edges with 10 pixels. Compared with the baseline, although our results slightly drop in "mean" and 30°, they are much better in "median" and 11.25°. It shows by preserving edge information, our geometry have more accurate predictions around boundaries. When evaluated over the entire images, our results outperforms the baseline in all the metrics, showing that our algorithm not only largely improves the prediction in planar regions, but also keeps the good predictions within non-planar regions.

**Depth criteria.** When evaluating depths, similarly, we also firstly adopt the traditional pixel-wise depth metrics that are defined in [7, 28]. We refer readers to the original papers for detailed definition due to limited space. We then also propose plane-wise metrics. Specifically, we generate the normals from the predicted depths using the NYU toolbox [24], and evaluate the degree variation (var.) of the generated normals within each plane.

| | | | Pixel-wise | | | | Higher the better | | Plane-wise |
| | | Lower the better (LTB) | | | | | | | LTB |
| Method | Rel | Rel(sqr) | $log_{10}$ | $RMSE_{lin}$ | $RMSE_{log}$ | 1.25 | $1.25^2$ | $1.25^3$ | var. |
|---|---|---|---|---|---|---|---|---|---|
| | | | | Evaluation over the planar regions | | | | | |
| Eigen-VGG [6] | 0.1441 | 0.0892 | 0.0635 | 0.5083 | 0.1968 | 78.7055 | 96.3516 | 99.3291 | 16.4460 |
| Post-Proc. | 0.1470 | 0.0937 | 0.0644 | 0.5200 | 0.2003 | 78.2290 | 96.1145 | 99.2258 | 11.1489 |
| Eigen-VGG(JT) | 0.1427 | 0.0881 | 0.0612 | 0.4900 | 0.1930 | 80.1163 | 96.4421 | 99.3029 | 17.5251 |
| DCRF | 0.1438 | 0.0893 | 0.0634 | 0.5100 | 0.1965 | 78.7311 | 96.3739 | 99.3321 | 12.0424 |
| DCRF(JT) | 0.1424 | 0.0874 | 0.0610 | 0.4873 | 0.1920 | 80.1800 | 96.5481 | 99.3326 | 10.5836 |
| DCRF-conf | 0.1437 | 0.0881 | 0.0631 | 0.5027 | 0.1957 | 78.9070 | 96.4336 | **99.3395** | 12.0420 |
| DCRF-conf(JT) | **0.1423** | **0.0874** | **0.0610** | **0.4874** | **0.1920** | **80.2453** | **96.5612** | 99.3229 | **10.5746** |
| Oracle | 0.1431 | 0.0879 | 0.0629 | 0.5043 | 0.1950 | 78.9777 | 96.4297 | 99.3605 | 8.0522 |
| Eigen-VGG [6] | 0.1645 | 0.1369 | 0.0735 | 0.7268 | 0.2275 | 72.9491 | 94.2890 | 98.6539 | Edge |
| DCRF-conf(JT) | **0.1624** | **0.1328** | **0.0707** | **0.6965** | **0.2214** | **74.7198** | **94.6927** | **98.7048** | |
| Eigen-VGG [6] | 0.1583 | 0.1213 | **0.0671** | **0.6388** | 0.2145 | **77.0536** | 95.0456 | 98.8140 | Image |
| DCRF-conf(JT) | **0.1555** | **0.1179** | 0.0672 | 0.6430 | **0.2139** | 76.8466 | **95.0946** | **98.8668** | |

Table 2: Depth accuracy comparison over the NYU v2 dataset.

**Evaluation on depth prediction.** Similarly, we first report the results on planar regions in the upper part of Tab. 2, and then present the evaluation on edge areas and over the entire image. We can observe similar trends of different methods as in normal evaluation, demonstrating the effectiveness of the proposed approach in both tasks.

**Qualitative results.** We also visually show an example to illustrate the improvements brought by our method. In Fig. 5, we visualize the predictions in 3D space in which the reconstructed strcture can be better observed. As can be seen, the results from network output [6] have lots of distortions in planar surfaces, and the transition is blurred accross plane boundaries, yielding non-satisfactory quality. Our results largely allievate such problems by incorporating plane and edge regularization, yielding visually much more satisfied results. Due to space limitation, we include more examples in the supplementary materials.

## 8 Conclusion

In this paper, we introduce SURGE, which is a system that induces surface regularization to depth and normal estimation from a single image. Specifically, we formulate the problem as DCRF which embeds surface affinity and depth normal compatibility into the regularization. Last but not the least, our DCRF is enabled to be jointly trained with CNN. From our experiments, we achieve promising results and show such regularization largely improves the quality of estimated depth and surface normal over planar regions, which is important for 3D editing applications.

**Acknowledgment.** This work is supported by the NSF Expedition for Visual Cortex on Silicon NSF award CCF-1317376 and the Army Research Office ARO 62250-CS.

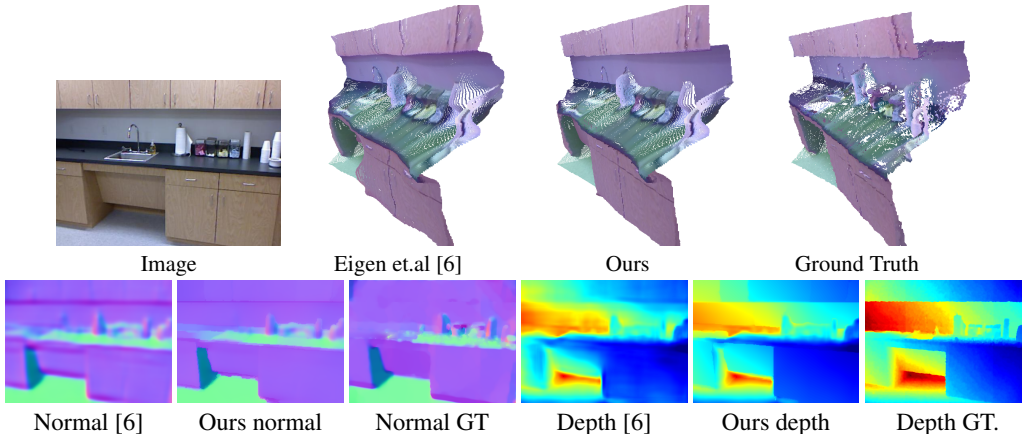

| Image | Eigen et.al [6] | Ours | Ground Truth |
| Normal [6] | Ours normal | Normal GT | Depth [6] | Ours depth | Depth GT. |

Figure 5: Visual comparison between network output from Eigen et.al [6] and our results in 3D view. We project the RGB and normal color to the 3D points (Best view in color).

## Footnotes

[1] http://cs.nyu.edu/~silberman/datasets/nyu_depth_v2.html

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
