[Supplementary Material]

# SURGE: Surface Regularized Geometry Estimation from a Single Image - Supplementary Material

**Peng Wang**[1]  **Xiaohui Shen**[2]  **Bryan Russell**[2]  **Scott Cohen**[2]  **Brian Price**[2]  **Alan Yuille**[3]

[1]University of California, Los Angeles  [2]Adobe Research  [3]Johns Hopkins University

**Content of this supplementary material.**

- Visualization of the confidence values of network prediction.
- Details of ground-truth planar surface extraction.
- Network architectures for plane and edge prediction.
- Learning and inference time.
- Additional qualitative results on the NYU v2 dataset.

Figure 1: Examples of the visualization of the prediction variation, which can be used to represent the prediction confidence by using the Monte Carlo dropout (last column). We highlighted the regions with large errors in terms of depth and normal prediction, and accordingly low confidence values.

# 1 Visualization of prediction confidence

In Fig. 1, we illustrate two examples of the confidence values used in our prediction for both depth and normal. We highlight the areas with large prediction errors, which are indicated by the low confidence values. We found that the normal confidence values are usually more accurate, while the depth confidence map is relatively more uniformly distributed with less variations. This is consistent with our experiments, which shows that incorporating normal confidence provides more performance gains, as shown in Tab.1 and Tab.2 respectively in the experiments in the main paper.

# 2 Details of extracting ground-truth planar surfaces

To extract the plane regions from the ground-truth depth data, we first obtain the planar surface candidates by processing the ground-truth depth maps using the plane extraction functions in the NYU toolbox [5], which is defined as minimum number (2500) of contiguous pixels that can be fit into a single 3D plane after projecting their depth values into the 3D world. We also obtain the regions of semantic instances from the semantic annotations provided in the dataset. For each semantic instance region $\mathbf{R}_i$, if we intersect it with the plane candidate regions, we can find a set of plane regions that are inside $\mathbf{R}_i$, which we denote as $\mathcal{S}_{\mathbf{R}_i} = \{\mathbf{P}_j | \mathbf{P}_j \subseteq \mathbf{R}_i\}$. If one of the regions in $\mathcal{S}_{\mathbf{R}_i}$ covers $90\%$ of the region of $\mathbf{R}_i$, it indicates that $\mathbf{R}_i$ should be mostly a planar region itself. In that case, we set $\mathcal{S}_{\mathbf{R}_i} = \{\mathbf{R}_i\}$ instead. We collect all such regions $\mathcal{S}_{\mathbf{R}_i}$ as the ground-truth planar surfaces in the image, i.e. $\mathcal{P} = \cup_i \mathcal{S}_{\mathbf{R}_i}$.

# 3 Network architectures for plane and edge prediction.

We present the details of the neural network architectures for edge and plane prediction respectively. Fig. 2 illustrates the architecture of the edge network . It has two streams. The bottom stream takes the RGB image as input, and has the same architecture as in HED [6], while the top stream is a very shallow one, which takes the predicted depth and normal maps as input to augment the prediction from images. In particular, it has a 64-channel convolutional layer with kernel size 9, followed by a fully-connected layer to generate the predictions. The predictions are then fuse into all the five side output in the HED stream.

Fig. 3 illustrates the architecture of planar surface network. It follows the multi-scale structure with side output from "conv3", "conv4" and "conv5" as in the fully convolutional network [4], but with a light-weighted network structure that is the same to the Deeplab-LargeFOV [2].

# 4 Learning and inference time

The inference includes performing 4-stream network forward propagation: 1.4s for depth and normal ($320 \times 240$ input), 1.5s for edge ($640 \times 480$) and plane ($320 \times 240$) with a single Titian X GPU, 10s for normalized cut with our Matlab implementation ($320 \times 240$ input), which could be substantially faster (0.2s) using GPU implementation from [1]. The final DCRF takes 1s for inference with a single CPU (3.2GHz). In total, it takes around 14s for each image ($640 \times 480$) in our current implementation. For training the networks, it takes around 10 hours each to train the edge and plane network, and takes around 2 days for fine-tuning over the depth and normal networks due to the CPU implementation of the DCRF layer.

Figure 2: Architecture of edge network with two input. At the corner of each layer we show the number of channels x the number of sub-layers.

Figure 3: Architecture of plane network that takes RGB image as input and generates a binary plane map.

# 5 Additional qualitative comparison from NYU v2 data.

In Fig. 4, Fig. 5, Fig. 6, we show more examples to visually demonstrate the improvement on the planar surface regions. For each example, we show an overhead 3D view at the first row, and corresponding normal and depth maps at the second row. The results from left to right are: Eigen et.al [3], ours and ground truth.

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

| Normal [3] | Ours normal | Normal GT | Depth [3] | Ours depth | Depth GT. |

Figure 4: Additional visual comparison between network output from Eigen et.al [3] and our results (Best view in color) from NYU v2 dataset.

| Normal [3] | Ours normal | Normal GT | Depth [3] | Ours depth | Depth GT. |

Figure 5: Additional visual comparison between network output from Eigen et.al [3] and our results (Best view in color) from NYU v2 dataset.

| Normal [3] | Ours normal | Normal GT | Depth [3] | Ours depth | Depth GT. |

Figure 6: Additional visual comparison between network output from Eigen et.al [3] and our results (Best view in color) from NYU v2 dataset.