[Reviews · NeurIPS 2016]

Reviewer 1

Summary

This paper proposes an approach to, given a single RGB image, estimating depth, normal for every pixel and predicting 3D planar information. Planar information is in turn used as a regularization term to improve the estimation of the depth and normal components. More specifically, off-the-shelf CNNs are independently user to predict four maps per image, namely: depth, normal, a planarity score and and edge map. These maps are then incorporated into a Dense CRF (DCRF). The gradient of the normal and the depth maps are back-projected to the original normal and depth CNNs allowing to retrain them. Several configurations of the proposed pipeline are evaluated and compared in terms of planar-wise metrics against Eigen [6], the original CNN used to estimate depth and normal maps. The framework proposed in this paper shows remarkable improved 3D maps, specially around the object boundaries and on the planar areas.

Qualitative Assessment

The paper is in general technically sound. Combining CNN outputs into a DCRF and back-projecting the errors of the DCRF into the CNNs is an interesting approach, already explored before for semantic image segmentation tasks [32]. Indeed, the authors mention they build upon the code made available in [32]. Additionally, the paper holds on the off-the-shelf CNNs of [6] and [31], for estimating each of the individual maps. All these previous contributions, especially [32], reduce the originality of the current submission. Nevertheless, I still appreciate bringing all these previous ideas to the topic of geometry estimation from single images. All ingredients are nicely combined, the definitions of the unary and pairwise potentials are interesting and can inspire the formulation of other types of geometric priors. Some points that need further clarification are the following: -It is not clear if the error of the CRF is propagated back through all the CNN and updates all parameters, or just the parameters of some of the layers. -The results are not very concluding. Since the proposed approach builds upon the CNN of [6], it is not surprising that the combination of this CNN with the Dense CRF improves the results of the original CNN [6]. Indeed, it would be interesting to compare the performance of the original [6] against the CNN that is re-trained with the DCRF error (but removing the DCRF layer). -In section 5.1, it is not clear how the mean-field approximation is combined with the updating rules of Eq.(6) and Eq.(7). A pseudocode of the process would clarify this. -The whole pipeline depends on the large number of hyperparameters that need to be empirically set. I wonder if they can be learned from data. -There are a number of grammatical errors and typos. Eg. Line 121 (K should be bold); Line 131 (z_i,z_j) should not be bold; Line 176 ((4). However); Line 224 (, and is used); Line 273 (our results outperform).

Confidence in this Review

2-Confident (read it all; understood it all reasonably well)


Reviewer 2

Summary

The paper a deep-learning based approach to estimate 3D-geometric information such as depth and surface normal from a single image with regularization using semantic edges and information about planes in the scene. It proposes a novel NN architecture with a 4-stream CNN to predict surface normals, depth, a planar mask and semantic edge respectively the output of which goes into a DCRF layer for non-local regularization. The paper also provides a back-propagation algorithm for end to end training. In the experiments, the paper further proposes a novel planar evaluation criteria and verify the robustness and effectiveness of their methods over the current state of the art on the NYU v2 depth dataset.

Qualitative Assessment

The paper provides a novel solution to depth estimation from a single image which significantly improves the estimation quality over the state of the art. In particular, the usage of planarity and semantic edges in this context is novel and is expected to have a significant impact on 3D geometry estimation in indoor scenes and other human-engineered places that have abundant planar surfaces as well as 3D editing applications for the same. The comparative results show clear value over the state of the art. Some suggestions: - A few more qualitative results in 3D views should be provided. - Please compare results with the work in reference [2]: A. Bansal, B. Russell, and A. Gupta., Marr Revisited: 2D-3D Model Alignment via Surface Normal Prediction, CVPR, 2016

Confidence in this Review

2-Confident (read it all; understood it all reasonably well)


Reviewer 3

Summary

See below.

Qualitative Assessment

The paper proposes a method for recovering scene geometry from a single RGB image. This method uses a dense CRF with terms that enforce consistency between point-wise depth and normal estimates, using regularizers based on classification of planarity and presence of depth boundaries. Each of these estimates (depth, normal, planarity, edges) comes from a separate network proposed for each task in prior work. Experimental evaluation show modest improvement over baseline depth and normal prediction models. In addition to the geometry-terms in proposed DCRF-based model, the paper's contributions include using multiple passes through the depth and normal networks with dropout to derive 'confidence' values of these metrics, and joint training to fine tune the depth and normal networks. While significantly engineered for its specific application domain, the paper does demonstrate a successful example of inference with a regularized objective, where different terms are predicted from trained neural networks. I believe this will be of interest to many in the NIPS audience. However, the overall presentation of the method---including some of the motivation and evaluation---leaves some questions un-answered: - I have to say the benefits from joint training don't appear to be significant---could this improvement be explained by essentially additional training iterations ? Perhaps, a better test would've been to train these networks from scratch, or perhaps further train the depth and normal networks for an equal number of further iterations on the same training set (but with direct losses instead of post-DCRF losses). - The fact that Eigen-VGG(JT) does better than the vanilla version is a little surprising---the joint training optimizes each network to perform better within the DCRF with complimentary information from the others. Why would it improve stand-alone performance ? Surely the direct loss used in the original training of each of these networks is the right choice for stand-alone performance. This again makes me believe that the improvement seen due to joint-training is because of the extra training iterations, and not because of the "joint" loss. - Why were confidences not used during joint training ? One could "tie together" multiple forward passes for the same example, and use that to derive confidence values, and backprop through these. Is it that the extra forward passes weren't worth the slower training time ? It is also worth reporting results for DCRF (JT)---i.e., DCRF results without confidence values, but with joint training---since this corresponds to using the network in the same regime for which it was trained. - While most of the terms in the DCRF objective "make sense", they are hand-crafted and involve making a lot of design decisions. Given the small size of the NYUv2 test set (and that its GT data is available), it is worth mentioning whether all these design decisions were made on the val set. Along similar lines, how did authors decide on the parametric form for dropout-variance to confidence ? Was this also determined on the val set from [9] ? - The work of [28] is perhaps the closest to the proposed method, since it also uses a dense CRF with joint terms for depth and edges. I think the paper could use a more thorough discussion of it. [28]'s results are worse than those reported in this paper, but that may be because of the VGG-trained [6] baseline network used by this paper. In fact, [28] appears to show a more significant relative improvement between its depth-only and depth+edge CRFs (than this paper's DCRF-conf(JT) vs [6]). A fair question to ask is how would [28]'s "HCRF" do with [6]'s depth network and VGG-pretraining. - The paper could do with a formal definition of 'planarity' means in the P0 output---specifically, at what resolution / region-size is point-wise planarity defined (in the limit, every infinitesimal surface point is a plane!) ? Is a planar region defined as one with some minimum number of contiguous pixels in which normals are constant ? - The baseline errors for [6] appear to be different from those reported in [6] itself. Could the authors comment ? (the biggest difference appears to be in the RMSE errors---are they computing means of mean(per-image-RMSE), or sqr-root(MSE across all images). I believe the latter is the convention used to report results on NYUv2. ===Post-rebuttal The authors have largely addressed the questions in my review. I'd say the paper just about meets the threshold for publication at NIPS (with the promised additional discussions for clarity). I'd also recommend that authors consider doing joint training with the RAW distribution of the NYUv2 set (rather than just the smaller official release). I think the current experiments might be underselling the utility of the method because of the limited training data.

Confidence in this Review

2-Confident (read it all; understood it all reasonably well)


Reviewer 4

Summary

This paper proposes a framework for geometry estimation, based upon CNN and dense CRF. In particular, the framework introduces regularizations that encourage planar surface structures. This regularization is formulated via dense CRF and is further coded as a trainable network layer. Furthermore, a novel planar-wise metric is proposed to evaluate geometry consistency within planar surfaces. Experimentally, qualitative and quantitive results both show that the proposed method significantly improves the state of the art.

Qualitative Assessment

I really like this work. Most existing loss functions for geometry estimation lacks enough respect to structures. For example, pixelwise difference has been widely used as the loss. In this work, a more structure-respecting metric is proposed, and the incorporation of this metric as a loss function for CNN is well executed, so that end-to-end training is possible.

Confidence in this Review

3-Expert (read the paper in detail, know the area, quite certain of my opinion)


Reviewer 5

Summary

This paper proposed a comprehensive neural network to jointly conduct single image normal and depth estimation. The novelty lies in incorporating high-order planar constraint and edge-constraint to generate more visually appealing results.

Qualitative Assessment

The author tackles the widely existing problems of most normal/depth estimation papers: the appealing depth/normal map estimation does not necessarily mean good 3D geometry estimation. In fact, small artifacts of of local normal map would result in very bad geometry in 3D view. That is mainly due to the lack of such constraint/objective when designing the network and loss functions. From this perspective I believe the author chooses the right direction and shows good results. The result outperforms the state-of-the-art results. The experimental design is solid. The paper is well-written in general. The demo in supp clearly show the advantage of the proposed method. But the technical novelty is somewhat limited. The idea of using boundary information has been explained in Origami World paper Fouhey et al. and combining all the cues including global scene structure have also been exploited in Wang et al., although the result in this submission is better. Incorporating DCRF into depth and normal estimation is also not new. The technical novelty lies in incorporating the comprehensive high-order geometric constraints into the DCRF + CNN framework. How to encode of all these cues into DCRF is well done. From application side, that is good enough. But I do not think the technical novelty is significant enough for a NIPS paper. Also I have doubts that such a comprehensive system might take a long time to run. Overall, given the good result and limited novelty I think this is a borderline paper.

Confidence in this Review

2-Confident (read it all; understood it all reasonably well)


Reviewer 6

Summary

This paper is about estimating geometry from a single image. The authors predict 3D planar surface information, and incorporate such information to generate better regularized geometry estimation. The method uses multiple CNNs to predict depth, normal, a binary plane map, and an edge map, then followed by a DCRF that fuses the four types of predictions.

Qualitative Assessment

This paper uses existing multiple CNNs to predict depth, normal, a binary plane map, and an edge map, then followed by a DCRF that fuses the four types of predictions. While all the components are not overall novel, the 3D planar regularization term is interesting. Previous works regularize the CRF in 2D, this paper proposes to do it in 3D, and shows improvement in experiments. Plane-wise metrics are also interesting, which evaluate the consistency of the predictions inside a ground truth planar region. The authors claim they do joint training of CNN and CRF. However, the parameters in CRF planar pairwise term are set by grid search on validation set. This is not joint training. For inference of depth and surface normal, the authors use coordinate decent optimization, which is not guaranteed to get converge.

Confidence in this Review

2-Confident (read it all; understood it all reasonably well)